# Neuroprotection and Beyond: The Central Role of CB1 and CB2 Receptors in Stroke Recovery

**DOI:** 10.3390/ijms242316728

**Published:** 2023-11-24

**Authors:** Bashir Bietar, Sophie Tanner, Christian Lehmann

**Affiliations:** 1Department of Pharmacology, Dalhousie University, Halifax, NS B3H 4R2, Canada; bashir.bietar@dal.ca (B.B.); sophietanner@dal.ca (S.T.); 2Department of Anesthesia, Pain Management, and Perioperative Medicine, Dalhousie University, Halifax, NS B3H 4R2, Canada

**Keywords:** endocannabinoid system, CB1 receptors, CB2 receptors, stroke recovery, neuroinflammation, immunomodulation, CIDS

## Abstract

The endocannabinoid system, with its intricate presence in numerous cells, tissues, and organs, offers a compelling avenue for therapeutic interventions. Central to this system are the cannabinoid receptors 1 and 2 (CB1R and CB2R), whose ubiquity can introduce complexities in targeted treatments due to their wide-ranging physiological influence. Injuries to the central nervous system (CNS), including strokes and traumatic brain injuries, induce localized pro-inflammatory immune responses, termed neuroinflammation. Research has shown that compensatory immunodepression usually follows, and these mechanisms might influence immunity, potentially affecting infection risks in patients. As traditional preventive treatments like antibiotics face challenges, the exploration of immunomodulatory therapies offers a promising alternative. This review delves into the potential neuroprotective roles of the cannabinoid receptors: CB1R’s involvement in mitigating excitotoxicity and CB2R’s dual role in promoting cell survival and anti-inflammatory responses. However, the potential of cannabinoids to reduce neuroinflammation must be weighed against the risk of exacerbating immunodepression. Though the endocannabinoid system promises numerous therapeutic benefits, understanding its multifaceted signaling mechanisms and outcomes remains a challenge.

## 1. Introduction

Approximately 15 million people worldwide suffer from stroke every year; this represents most of the central nervous system (CNS)-related injuries [1]. The most frequent etiology of acute stroke is represented by a thrombus occluding an artery supplying the brain (i.e., ischemic stroke), causing hypoxia and subsequent death of brain tissue [2]. At the cellular level, the lack of oxygen supply causes failure in the production of the energy-carrying molecule, ATP. This shifts cells to anaerobic glycolysis, leading to the accumulation of lactic acid, which in turn causes lactic acidosis [3]. Voltage-dependent calcium and sodium channels are activated, and high calcium levels in the cells lead to catabolic intracellular processes that damage essential membranes and proteins [4]. On the other hand, the disproportionate increase in sodium levels causes an increase in water influx, and as a result, the cells swell, causing edema. The disproportionate sodium and calcium levels in ischemic cells incites a pathologic increase in the production of reactive oxygen species (ROS), which directly destroys proteins, lipids, carbohydrates, and nucleic acids [5]. The product of these cellular events is a mixture of neuronal and vascular cell death via necrosis, apoptosis, and autophagy [6].

Consequentially, the cellular pathophysiology induces a sequence of immune events, starting with the release of inflammatory mediators and the migration of leukocytes to the site of injury. This local inflammatory response is governed by the microglia, which are resident immune cells of the CNS that can detect pathologic changes and respond by exhibiting characteristic macrophage functions such as cytokine release (mainly, tumor necrosis factor (TNF), interleukin-1β (IL-1β), and interleukin-6 (IL-6)), phagocytosis, and antigen presentation [7]. The aim of this initial immune response is to recruit immune cells and clean up cell detritus. However, this acute inflammatory state of the CNS (i.e., neuroinflammation) is not supposed to persist, and the immune system attempts to restore homeostasis via the release of anti-inflammatory mediators such as IL-10 while simultaneously curbing the release of pro-inflammatory mediators. This compensatory immunodepression (i.e., CNS injury-induced immunodepression syndrome—CIDS) is not restricted to the brain and can increase susceptibility to peripheral infections, causing excess morbidity and mortality [8,9]. Patients with CNS injuries are more susceptible to hospital-acquired infections, particularly nosocomial pneumonia. Research indicates that CNS injury activates microglia, with this change being associated with acute lung injury in neurotrauma rats [10,11].

The complexities of CIDS hamper the development of effective, tissue-damage mitigating therapies and present considerable challenges for new drug development. Preclinical and clinical evidence clearly demonstrate the therapeutic potential of regulating neuroinflammation. Thus, immunomodulation has garnered the interest of researchers in the field. The intricate mechanisms of neuroprotection and stroke recovery continue to be a pivotal focus in medical research. As we delve into the cellular and molecular intricacies underlying these processes, the significance of interrelated biological systems becomes increasingly apparent. One such system, drawing heightened attention for its multifaceted role in neurology, is the endocannabinoid system (ECS). The ECS, with its extensive involvement in neural functions and its potential therapeutic implications, offers a compelling avenue for exploration. For example, Eitan et al. (2023) recently explored the impact of the synthetic cannabinoid XLR-11, a non-selective cannabinoid receptor agonist. Their study revealed that treatment with XLR-11 in human brain microvascular endothelial cells (hBMVEs) led to increased expression of key angiogenic factors such as vascular endothelial growth factor (VEGF), angiopoietin-1 (ANG1), and angiopoietin-2 (ANG2). The elevation of these factors implies a role of ECS activation in improving stroke outcome [12]. Angiogenesis is only one reported benefit among many other beneficial effects of ECS modulation. This leads us to a detailed examination of the ECS, particularly focusing on the selective activation of its receptors and their roles in neuroprotection, anti-inflammation, and stroke recovery.

## 2. Endocannabinoid System

The endogenous cannabinoid system (ECS) is a lipid signaling system that has essential regulatory functions throughout the body in all vertebrates. The main endocannabinoids are small molecules derived from arachidonic acid, namely, anandamide (AEA) and 2-arachidonoylglycerol (2-AG) [13,14]. These molecules interact with G-protein-coupled receptors (GPCRs)—specifically, the cannabinoid receptors (CBRs), CB1R and CB2R, which are both part of the Class A (Rhodopsin family) of GPCRs [15,16]. Both CB1Rs and CB2Rs exhibit seven transmembrane domains, an extracellular N-terminus, and an intracellular C-terminus. Despite being 44% homologous overall and 68% homologous within the transmembrane regions, there have been natural polymorphisms identified in these receptors [17,18,19].

The signaling pathways associated with CB1/2R are mediated by Gi/o proteins, part of the GPCR receptor family. Upon activation, CB1/2Rs suppress adenylyl cyclase and cyclic adenosine monophosphate (cAMP) [20], promote mitogen-activated protein kinase (MAPK) signaling, stimulate phosphoinositide 3-kinase (PI3K) pathways, and additionally modulate voltage-gated calcium channels [21,22]. Moreover, CB1/2Rs can be phosphorylated by G-protein-coupled receptor kinases (GRKs) and then associate with β-arrestin1 or β-arrestin2, which influence receptor internalization and encourage interactions with proteins that steer signaling along β-arrestin-mediated pathways [23].

Located on the cell surface and within internal structures such as endosomes, the endoplasmic reticulum, and the mitochondrial membrane, CB1/2Rs change the inactive guanine nucleotide GDP to its active form, GTP, upon activation [24,25]. This process facilitates the dissociation of the heterotrimeric G protein into α and βγ subunits. Each of these subunits regulates distinct pathways. The βγ subunits manage phospholipase C (PLC) isoforms and stimulate the MAPK signaling network [26], and the α subunit (Gαi) attaches to and inhibits adenylate cyclase, subsequently controlling the synthesis of the second messenger, cAMP, and governing downstream cAMP-dependent signaling events [27].

Ligands interacting with CB1/2R can be classified into orthosteric and allosteric, depending on their binding method or location on the receptor. Orthosteric ligands bind at the same location as the endogenous agonist, whereas allosteric ligands connect at different receptor sites. Pure allosteric modulators solely refine the action of the orthosteric ligand by adjusting the conformation of the receptor protein and influencing the binding affinity and/or efficacy of orthosteric ligands [28].

Among common naturally occurring cannabinoids, both CB1R and CB2R are activated by the endocannabinoids AEA and 2-AG and the phytocannabinoid Δ9-tetrahydrocannabinol (THC). Synthetic cannabinoids like CP 55,940 and WIN 55,212-2, which target CB1/2R, have been identified as potent but non-selective CBR ligands due to their comparable efficiency in activating both CB1R and CB2R [17]. As one would expect, numerous selective agonists and antagonists were rapidly developed and made available to the scientific community. Several CB1R-selective agonists and antagonists have been developed and made available for research purposes, including the agonists ACEA, ACPA, and arachidonylcyclopropylamide, as well as the inverse agonists AM251 and rimonabant [29,30,31]. CB2R-selective ligands, including the agonists HU 308, JWH 133, L-759633, L-759656, and AM1241 and the inverse agonists SR144528 and AM630, have also been formulated [32,33].

In summary, the ECS plays a crucial role in regulating various bodily functions in all vertebrates through its lipid-based signaling system. The ECS utilizes endocannabinoids and synthetic cannabinoids, which interact with specific receptors and stimulate different signaling pathways to exert their regulatory effects. Our understanding of these mechanisms continues to grow, aided by the development of various selective agonists and antagonists that target specific receptors. Further research into this complex system is expected to provide critical insights into its roles and potential therapeutic applications, opening new avenues in the field of medical science. 

## 3. Cannabinoid Receptor 1 (CB1R) in Neuroprotection: Mitigating Excitotoxicity

The CB1R was identified by Devane et al. in 1988 and was found in the CNS, particularly in brain regions responsible for memory, emotion, and motor control. CB1R is encoded by the CNR1 gene and is the most abundantly expressed among 100 G-protein-coupled receptors in the brain [13]. It is chiefly expressed in the cortex, hippocampus, and basal ganglia nuclei. As a result of its distribution within the CNS, CB1R (responsible for the psychoactive effects of cannabis inhalation) is heavily distributed in areas of the brain concerned with motor function, cognition, emotion, and homeostasis [16].

CB1Rs are mainly located on the presynaptic nerve terminals and are involved in modulating neurotransmitter release, and their activation can affect synaptic transmission. CB1R signaling plays a crucial role in maintaining neural homeostasis through the modulation of cAMP production and the activation of MAPK signaling cascades. For instance, in neuronal cells, stimulation of CB1R inhibits adenylyl cyclase and reduces cAMP levels, which in turn modulates the activity of downstream effectors such as protein kinase A (PKA), ultimately influencing synaptic transmission and plasticity [34,35]. In hippocampal neurons, CB1R agonists induce extracellular signal-regulated kinase 1/2 (ERK1/2) activation in a Gi/o-protein-dependent manner, independently of adenylyl cyclase inhibition. This activation of ERK1/2 has been reported in various neuronal cell types, highlighting its physiological relevance in the nervous system [35,36].

In a study by Chou et al., the researchers investigated the distribution of CB1R in the human and non-human primate prefrontal cortex and auditory cortices. Using co-labeling immunohistochemistry and fluorescent microscopy, they examined CB1R protein levels within excitatory and inhibitory boutons, which are crucial for understanding the receptor’s role in regulating excitatory and inhibitory (E/I) balance [37,38]. The study found that CB1R was present in both excitatory and inhibitory boutons in all examined brain regions of both species. However, inhibitory boutons showed significantly higher CB1R levels compared to excitatory boutons across all regions [37]. CB1R exists in presynaptic neurons of (gamma-aminobutyric acid) GABA-ergic axon terminals [18,39,40]. It negatively regulates neurotransmitter release through the inhibition of excitatory neurotransmitter release by regulating A-type potassium channel and N-type calcium channel terminals [18,39,40]. This effect has been shown to play a role in pain modulation, where CB1R activation inhibits the release of pro-inflammatory mediators and decreases the perception of pain [41,42].

Stroke involves the over-release of excitatory neurotransmitters such as glutamate, which can lead to excitotoxicity—a process involving intracellular calcium overload, the activation of harmful enzymes, and oxidative stress, ultimately resulting in neuronal injury or death [43,44]. However, the ECS can offer protection against excitotoxicity. The activation of CB1R can inhibit the excessive release of glutamate, thereby reducing excitotoxicity and neuronal damage [45]. In particular, activation of CB1R has been associated with neuroprotective effects in animal models of neurodegenerative diseases, including Parkinson’s disease, Alzheimer’s disease, and Huntington’s disease [46].

A study aimed to evaluate the neuroprotective effect of an acute activation of CB1R and its impact on inducible nitric oxide synthase (iNOS) protein expression, nitric oxide (NO) levels, gliosis, and the neurodegenerative process induced by the injection of Aβ(25–35) into the CA1 subfield of the hippocampus. The results revealed that the administration of a CB1R agonist prevented both the increase in iNOS protein and NO levels and the reactive gliosis induced by Aβ (25–35) and significantly reduced neurodegeneration in the CA1 subfield of the hippocampus [47].

Another study reported on the functional interaction of metabotropic glutamate receptor 5 (mGluR5) and CB1R in promoting neuroprotection. CB1R regulates pre-synaptic glutamate release, and mGluR5 activation increases endocannabinoid synthesis at the post-synaptic site. Using primary cultured corticostriatal neurons, the researchers showed that either the pharmacological blockade or the genetic ablation of either mGluR5 or CB1R can abrogate both CB1R- and mGluR5-mediated neuroprotection against glutamate insult (excitotoxicity) [48]. These two receptors work cooperatively to trigger the activation of cell signaling pathways to promote neuronal survival, which involves MEK/ERK1/2 and PI3K/AKT activation. Intact signaling of both receptors is required to effectively promote neuronal survival. In conclusion, mGluR5 and CB1R cooperate to confer neuroprotection by activating certain cell signaling pathways and promoting neuronal survival [48].

The neuroprotective activity of CB1R is closely linked to its ability to inhibit excitotoxic neurotransmission by blunting pre-synaptic glutamate release. Additionally, there is a strong association between CB1R activity and the expression of brain-derived neurotrophic factor (BDNF), a master neurotrophin in the mammalian forebrain. This connection is particularly relevant in the striatum, where CB1R and BDNF, along with their high-affinity receptor TrkB, play vital roles in the survival, generation, and plasticity of medium-sized spiny neurons. The decline in expression of these elements in animal models of neurodegenerative diseases like Huntington’s disease (HD) and Parkinson’s disease and their restoration prevents HD-like neurodegeneration. Intriguingly, experiments in the R6/2 mouse model of HD showed that enforced re-expression of the CB1 receptor in the dorsolateral striatum allowed the re-expression of BDNF and helped rescue neuropathological deficits [49].

Further supporting the role of CB1R in neuroprotection, a study on pulsed electromagnetic fields (PEMF) yielded intriguing findings. The research demonstrated that PEMF exposure promotes survival of HT22 neuronal cells subjected to excitotoxic conditions, reduces lactate dehydrogenase release (an indicator of cellular damage), and decreases cell death. CB1R’s role was highlighted when its antagonists (AM251 and AM281) negated the protective effects of PEMF. Notably, increased levels of anandamide and 2-AG were observed post PEMF exposure. This suggests that the neuroprotective effects of PEMF might be facilitated by the modulation of the endocannabinoid (eCB) metabolic system. Importantly, the study showed reduced expression of monoacylglycerol lipase (MAGL) and fatty acid amide hydrolase (FAAH) following PEMF, suggesting diminished endocannabinoid hydrolyzation and, consequently, increased endocannabinoid levels. The study further underscored the CB1R/ERK signaling pathway as a crucial mechanism through which PEMF modulates excitotoxicity, as confirmed by using AM251 and ERK 1/2 inhibitors. Overall, this research broadens our understanding of CB1R-associated neuroprotection, suggesting that shielding neurons from excitotoxicity can occur through a mechanism involving the eCB/CB1R/ERK signaling pathway [50].

More important to our discussion, however, is CB1R’s role in cerebral ischemia models. Spyridakos et al. found that acute and subchronic administration of WIN-55212-2 induced neuroprotection and anti-inflammatory actions in rat retina. This effect was reported to be due to CB1R activation [51]. In support of this, the administration of WIN 55.212-2 (through CB1R activity) was found to reduce neurological damage and infarct size in rat models of artery occlusion and minimized damage after hypoxic-ischemic brain injury in preterm lambs [52,53]. Further confirming the role of CB1R, Jia et al. found that anandamide protects HT22 cells exposed to hydrogen peroxide by inhibiting CB1 receptor-mediated type 2 NADPH oxidase. Panikashvili et al. (2001, 2005, and 2006) and Mecha et al. (2019) studied the effects of 2-AG. Mecha et al. found that 2-AG enhances spontaneous remyelination by targeting microglia, and Panikashvili and colleagues found that 2-AG is neuroprotective after brain injury, protects the blood–brain barrier after closed head injury, and inhibits mRNA expression of proinflammatory cytokines. These were all reported to be due to 2-AG’s activity on CB1R [54,55,56,57,58].

Preclinical studies of selective CB1R agonists, such as ACEA, have shown promising results in alleviating injuries and improving neurological outcomes in C57Bl/6 mice subjected to middle cerebral artery occlusion (MCAO). The mice received CB1 receptor agonist ACEA, CB1 receptor antagonist AM251, or vehicle at different time points after MCAO. ACEA treatment reduced astrocytic reactions, neuronal death, and dendritic loss, whereas AM251 treatment increased these parameters. Motor tests revealed that only ACEA treatment countered the progressive deterioration in motor activity observed in ischemic animals [59]. Andres-Mach et al. (2015) and Liu et al. (2022) both studied the effects of ACEA in mouse models. The former found that ACEA stimulated hippocampal neurogenesis in mice treated with antiepileptic drugs, and the latter found that ACEA attenuates oxidative stress by promoting mitophagy via the CB1R/ nuclear respiratory factor 1 / PTEN-induced kinase 1 pathway after subarachnoid hemorrhage [60,61].

In contrast, however, two studies showed that CB1R antagonism was instead beneficial. The first study investigated the impact of CB1R activation during ischemia on stress and reward signaling, neuronal injury, and anxiety-like behavior in male Wistar rats. The rats were divided into four groups, and CB1R antagonist AM251 was administered prior to ischemia induction. The findings showed that AM251 reduced CA1 injury and behavioral deficits in ischemic rats, normalized dopamine-related markers, and diminished anxiety-like behaviors [62]. Another antagonist, SR141716A, reduced infarct size and stroke volume when administered both as a pre-treatment and post-treatment [63].

Though ischemia typically triggers cell death pathways, it also activates innate cell survival pathways, yet their precise nature remains elusive. Intriguingly, CB1R’s role extends to astroglial cells. Research using the selective endocannabinoid clearance enzyme inhibitor JZL195 in a male mouse model of global cerebral ischemia showed that JZL195 administration induced sustained specific synaptic changes that promoted neuroprotection [64]. These synaptic modifications depended primarily on CB1R activation in astroglial cells, with additional involvement from glutamatergic or GABAergic neurons. Key players in this process were glutamate NMDA (N-methyl-D-aspartate) receptors and synaptic trafficking of glutamate AMPA (α-amino-3-hydroxy-5-methyl-4-isoxazolepropionic acid) receptors. In essence, enhancing extracellular endocannabinoid levels through inhibiting their clearance leads to neuroprotection via long-term synaptic depression, mediated by sequential CB1R activation in astroglia and subsequent engagement of postsynaptic glutamate receptors [64].

In summary, CB1R-mediated neuroprotection in stroke appears to involve a series of interconnected mechanisms. First and foremost, CB1R activation inhibits excitotoxicity by curtailing excessive glutamate release, a major contributor to neuronal injury and death in stroke, working in tandem with metabotropic glutamate receptor 5 (mGluR5) to safeguard against glutamate-induced toxicity, suggesting a role in preserving synaptic function. Excitotoxicity is considered a critical factor in the progression from ischemia to neuronal death. This understanding has led to clinical trials testing various inhibitors of excitotoxicity in stroke patients. Glutamate, primarily acting through the calcium-permeable ionotropic NMDA receptor, plays a significant role in excitotoxicity. Subpopulations of the NMDA receptor can generate different functional outputs, influencing the progression of neuronal damage during ischemic events [65]. These trials aimed to intervene in the mechanistic steps leading to excitotoxicity to prevent stroke damage. Clinical trials of drugs targeting NMDA receptors for stroke treatment have generally failed to show positive outcomes, often worsening mortality and morbidity instead of improving them. Drugs like Selfotel, Midafotel, Aptiganel, and Gavestinel, though theoretically promising, were plagued by significant side effects such as agitation, confusion, and hypertension, attributed to the widespread neurological functions of NMDARs [66]. These challenges, coupled with a narrow therapeutic window for effective treatment, have led to the discontinuation of many of these drugs from further clinical development. CB1R modulation presents a promising alternative for stroke treatment due to its distinct mechanism of action and potential for reduced side effects compared to NMDA receptor-targeting drugs. Furthermore, CB1R modulation has been linked to the promotion of neurogenesis and the expression of neuroprotective factors like BDNF, enhancing the brain’s natural recovery processes. This targeted approach, focused on endocannabinoid system components, offers a more nuanced method to mitigate stroke damage, potentially leading to more effective and safer stroke therapies. CB1R also helps manage the inflammatory response after a stroke, as evidenced by the CB1R agonist ACEA reducing astrocytic reaction and oxidative stress, both key markers of neuroinflammation. Lastly, CB1R is implicated in activating cell signaling pathways that are crucial for neuronal survival, including MEK/ERK1/2 and PI3K/AKT pathways, as demonstrated in studies on pulsed electromagnetic fields. Thus, CB1R-mediated neuroprotection seems to take a comprehensive approach, involving glutamate regulation, inflammation control, promotion of neuronal survival, and synaptic preservation, ultimately salvaging tissue during ischemic events and reducing the impact of stroke. Table 1 contains a summary of existing research.

## 4. The Role of Cannabinoid Receptor 2 (CB2R) in Neuroprotection: Cell Survival and Anti-Inflammation

In addition to its functions in the brain, the endocannabinoid system is an important modulator of the immune system via CB2R activation. The CB2R was initially identified on the surface of macrophages in the spleen by Sean Munro in 1993 using PCR [19]. CB2R is encoded by the Cnr2 gene. The pattern of CB2R expression among human tissues is consistent between studies. Measuring receptor mRNA levels using PCR showed that CB2Rs are predominantly expressed in bones and in peripheral immune organs, such as the thymus, spleen, and tonsils, at varying levels [77,78]. Furthermore, the high level of CB2R expression in human immune tissues was recapitulated in murine and rodent models, and similar expression patterns were observed [18]. CB2Rs have also been found in the male and female reproductive systems, the cardiovascular system, and the respiratory system [79]. Although it was previously thought that CB2Rs were exclusively peripheral receptors, recent studies showed that they also exist in the CNS [80]. Researchers investigated the expression of CB2 cannabinoid receptors in the brain, which has been less well established compared to CB1 receptors. The presence of CB2Rs on neurons has been a subject of debate, but evidence demonstrates their expression on brainstem and midbrain dopaminergic neurons [81,82]. Evidence showed that CB2Rs were expressed in neurons, glial cells, and endothelial cells. In neuronal cells, CB2R was observed in somata and large- and medium-sized dendrites. In the soma, CB2R labeling was primarily associated with the rough endoplasmic reticulum and Golgi apparatus, suggesting endogenous synthesis. In the dendrites, CB2R was observed in the cytoplasm and near the plasma membrane at synaptic contact areas with axon terminals, indicating a postsynaptic distribution of these receptors [81]. In the substantia nigra, some unmyelinated axons were immunoreactive for CB2Rs, but CB2R-labeled axon terminals were rare. These findings indicate that CB2Rs are primarily postsynaptic in the CA1 area of the hippocampus and the substantia nigra [81,82,83]. Of note, however, neuronal CB2Rs display lower baseline expression in comparison to CB1Rs, yet they exhibit a heightened responsiveness to specific stimuli, such as inflammation [83,84,85]. Evidence shows that CB2R expression is upregulated during brain inflammation or disease, for example, in multiple sclerosis and Alzheimer’s disease, suggesting a link between CB2R expression and immune cell activation [25,46,86]. CB2R immunofluorescence was increased on GFAP-positive astrocytes and Iba1-positive microglia in the hippocampus and cortex of 17-month-old arcAβ mice compared to non-transgenic littermate (NTL) mice. Additionally, CB2R immunofluorescence was higher in glial cells inside amyloid-β deposits than peri-plaque glial cells [86]. These findings suggest specific increases in microglial and astroglial CB2R expression levels during neuroinflammation, emphasizing its potential as a target.

In a stroke model of cerebellar lesions in one cerebellar hemisphere, remote cell death and the increased expression of CB2R were triggered in contralateral precerebellar neurons. Through a series of selective agonist and antagonist interventions that modulated cannabinoid receptor activity, it was discovered that the CB2R agonist JWH-015 exerted a reduction in neuronal loss and cytochrome-c release, ultimately contributing to neurological recovery [71]. Collectively, Viscomi et al. provide compelling evidence that axonal damage triggers CB2R expression in central neurons, and subsequent stimulation of CB2R elicits a neuroprotective effect primarily mediated through PI3K/Akt signaling [71]. This pathway is widely reported to be involved in neuroprotection [87,88]. Bravo-Ferrer et al. utilized a murine model of permanent MCAO to induce stroke. They examined CB2R modulation by administering both the CB2R agonist JWH133 and the CB2R antagonist SR144528 over specific time periods. Additionally, they investigated the function of CB2R through genetic deletion. The administration of the CB2R antagonist SR144528 was associated with a decrease in neuroblast migration and new neuron formation following stroke. In contrast, JWH133 did not yield significant impacts on neurogenesis or stroke outcomes in vivo. However, it did enhance the migration of neural progenitor cells in vitro. These results emphasize the complex influence of CB2R on neuroblast migration and, consequently, neurogenesis in the post-stroke context [89].

Upon neuroinflammation, microglia become activated and further increase CB2R expression. Similarly, in response to stroke, microglia transform into activated states. Activated microglia can release various pro-inflammatory cytokines such as TNF, IL-1β, and IL-6, contributing to inflammation-related neuronal damage [90]. CB2Rs are mainly expressed on immune cells, including microglia within the brain. Researchers have shown that CB2Rs are expressed by microglia in the brain, and the stimulation of CB2R in the BV2 microglial cell line was shown to inhibit adenylyl cyclase and its downstream effectors such as phosphorylated cAMP-dependent protein kinase (p-PKA), leading to M1/M2 polarization and neuroprotection [91,92]. This represents a shift towards an anti-inflammatory state. Increasing evidence shows the neuroprotective role of CB2R agonism following stroke [93]. The anti-inflammatory state in the brain allows for the establishment of a regulatory phase that can potentially control neuroinflammation. Eljaschewitsch et al. demonstrated that CB2 receptor stimulation prompts inhibitory signals within microglia, achieved through the inhibition of the cAMP/PKA pathway and the suppression of transcription factors NF-AT (nuclear factor of activated T-cells) and AP-1 (activator protein- 1). In the context of their investigation, the authors specifically demonstrated that the endocannabinoid AEA instigates histone H3 phosphorylation, MAPK phosphatase-1 (MKP-1) expression, and consequent ERK-1/2 dephosphorylation in activated microglial cells. This intricate process effectively suppresses the release of NO, culminating in a state of neuroprotection [54,94].

In a study using MCAO, Yu et al. found that the expression of CB2R, along with microglial inflammatory markers IBA1 (ionized calcium binding adaptor molecule 1) and TLR4 (toll-like receptor 4), significantly increased after stroke, peaking on the 5th day. This upregulation of CB2R occurred much earlier and was more prominent than that of IBA1 or TLR4. The stroke-induced rats were treated with the CB2R agonist AM1241 or the anti-inflammatory agent pioglitazone from days 2 to 5 after the stroke. Post-treatment with pioglitazone significantly reduced brain infarction, neurological score, and IBA1 expression, suggesting a neuroprotective effect through the suppression of inflammation. Though AM1241 slightly reduced CD4 and CD8 lymphocyte infiltration in the peri-lesioned area, it failed to alter brain infarction, neurological deficits, and IBA1 mRNA expression in stroke brain. This suggests that delayed post-stroke treatment with CB2R agonist AM1241 cannot efficiently suppress microglial activation, brain damages, or behavioral deficits in stroke animals. However, when AM1241 was administered 5 min before the stroke, it significantly reduced the area of infarction and neurological symptoms in stroke rats [95]. This suggests that the neuroprotective effects of CB2R agonists in stroke are time-dependent, with early treatment suppressing neurodegeneration and inflammation. Figure 1 shows the importance of a timely intervention in the context of stroke.

In a study, male mice were subjected to transient MCAO, and two distinct selective CB2R agonists (O-3853, O-1966) were intravenously administered—one hour prior to MCAO or ten minutes after reperfusion. Leukocyte/endothelial interactions were assessed across three time points: pre-MCAO, 1 h post MCAO, and 24 h post MCAO, utilizing a sealed cranial window approach. Cerebral infarct volume and motor function were subsequently measured 24 h post MCAO. The administration of selective CB2 agonists yielded significant reductions in cerebral infarction and tangible enhancements in motor function following 1 h of MCAO, succeeded by 23 h of reperfusion in mice. Importantly, ischemic untreated animals exhibited significant increases in leukocyte rolling and adhesion in both venules and arterioles. However, rolling and adhesion were significantly reduced by the application of selective CB2R agonists, whether administered one hour prior to or ten minutes after MCAO. Collectively, these findings support a connection between CB2R activation, a reduction in leukocyte rolling and adhesion along cerebral vascular endothelial cells, and diminished infarct size and improved motor function after transient focal ischemia [96,97].

Furthermore, CB2R stimulation can enhance the production of anti-inflammatory cytokines like interleukin-10 (IL-10). AEA has been observed to enhance IL-10 production in microglial cells by activating CB2R, primarily through the mediation of ERK1/2 and JNK MAPKs [98]. Notably, endogenously produced IL-10 exerts a regulatory influence on interleukin- 12 (IL-12) and interleukin-23 (IL-23) cytokines, consequently affecting the transcription factor profiles associated with T-helper cell commitment in splenocytes [99,100]. These findings imply that CB2 activation may influence the nature of immune responses within the CNS while still impacting splenocytes. This presents an opportunity for cannabinoid-mediated intervention against neuroinflammation and neurodegeneration.

In summary, studies show that the activation of CB2 receptors plays a pivotal role in safeguarding the brain by ameliorating cerebral microcirculatory dysfunction, neuroinflammation, and deleterious effects that occur during cerebral ischemia/reperfusion injuries. Furthermore, CB2R is implicated in cell survival pathway activation, and its knockout has been shown to crucially impact neurogenesis and cell survival. The overall effect of CB2R activation is a shift towards a more neuroprotective environment, decreasing neuronal injury and promoting recovery after stroke. This shift is often attributed to M1/M2 polarization, with M2 microglia being responsible for the anti-inflammatory/reparative environment. Once microglia have switched, cytokine release and leukocyte recruitment, adhesion, and function are all influenced by the action of M2 microglia. The microenvironment created then can preferentially favor anti-inflammation and neuroprotection, thus decreasing stroke size and the related motor deficits. Cell survival pathways have been speculated to be regulated by CB2R and to play a role in neuroprotection, and as such, studies have aimed to unravel the mechanisms underlying the neuroprotective properties of cannabinoids both in vivo and in vitro (Table 1). In a large randomized clinical trial involving 1216 patients with acute ischemic stroke, DL-3-n-butylphthalide (NBP), a neuroprotective agent already approved since 2002 for ischemic stroke treatment in China, significantly improved functional outcomes at 90 days compared to a placebo when used alongside reperfusion therapies like intravenous thrombolysis and endovascular treatment. The trial, conducted across 59 centers in China, found that 56.7% of patients in the NBP group achieved favorable outcomes based on the 90-day modified Rankin Scale score, compared to 44.0% in the placebo group, without a notable increase in serious adverse events [101]. NBP’s mechanism of action utilizes similar pathways as CB2R agonism. Namely, NBP has been shown to reduce oxidative stress, reduce neuronal cell apoptosis through PI3K/Akt signaling, and improve neurogenesis [102].

In a phase III clinical trial, patients treated with edaravone dexborneol within 48 h of AIS onset showed better functional outcomes at 90 days compared to those receiving edaravone alone, particularly in female patients [103]. The mechanism of action of this combination involves edaravone acting as a free radical scavenger, reducing oxidative stress and neuronal damage associated with stroke. Dexborneol contributes by enhancing the permeability of the blood–brain barrier, allowing for more efficient drug delivery to affected brain tissues. Additionally, dexborneol exhibits anti-inflammatory properties and may protect against cell death pathways. This study highlighted the importance of a narrow therapeutic window when treating ischemic stroke. The optimal time window is believed to be within 12–48 h of injury [103]. This parallels the findings of time-dependent improvement in stroke outcome using the CB2R agonist AM1241.

Put in this context, CB2R modulation can be beneficial in the early post-stroke period by acting on varying cytoprotective pathways. Despite the acknowledged neuroprotective capabilities of cannabinoids, the precise mechanisms orchestrating their effects have remained largely unexplored. This uncharted territory is where potential risks of cannabinoid-mediated neuroprotection may exist. Namely, the wanted effect of anti-inflammation and neuroprotection can bring about exacerbated immunodepression due to CNS injury-induced immunodepression syndrome.

## 5. Anti-Inflammation That Can Exacerbate Immunodepression

Experimental data suggest that CB2R activation is beneficial mainly in the early post-stroke phase. Less is known about the late consequences of CB2R activation, particularly in the periphery, after brain injury. More specifically, anti-inflammation caused by peripheral CB2R activation could contribute to CIDS and negatively impact outcomes of patients after an acute CNS injury, leaving them susceptible to infections.

Gene expression studies indicate that the CB2R mRNA is expressed by most cell types of the rodent and human immune system. It is useful, then, to determine the identity of cells expressing CB2R protein. Detailed flow cytometry analysis of CB2R protein levels expressed by blood-derived immune cells from healthy human donors revealed that NK cells, B-lymphocytes, and monocytes expressed a higher level of CB2R than T-lymphocytes (CD4+ or CD8+) and neutrophils [104]. NK cells showed the most variation in CB2R expression levels, whereas the other cell types were reported to have relatively similar CB2R expression levels between subjects [104]. Moreover, the most reported outcomes of CB2R activation in immune cells include the inhibition of proliferation, the induction of apoptosis, effects on cytokine/chemokine production, and leukocyte adhesion and migration. In human PBMCs, studies indicate that CB2R activation inhibits interleukin-2 (IL-2), interleukin-17 (IL-17), interferon-gamma (IFN-γ), and TNF secretion and stimulates IL-4, IL-10, and transforming growth factor beta (TGF-β) secretion [105]. This cytokine profile suggests a largely anti-inflammatory effect of CB2R activation. CB2R-induced cAMP inhibition on murine-derived splenic T-cells was largely anti-inflammatory, with a suppression in the release of IL-2, the main cytokine responsible for T-cell proliferation. AEA has been shown to reduce mitogen-induced T-cell proliferation and induce apoptosis. Furthermore, a reduction in IL-2 synthesis can be mediated by both CB1R and CB2R activation by AEA, which impacts T-cell proliferation and may thus be an undesirable artifact of CBR agonism [106,107]. Interestingly, prolonged exposure to the CB2R agonist made T-cells become insensitive to inhibition by cannabinoids, because the CB2R receptor becomes downregulated [108].

Activation of CB1R and CB2R on T-cells has been found to reduce their migration, decrease proliferation, inhibit the secretion of pro-inflammatory cytokines, and promote the secretion of anti-inflammatory cytokines. 2-AG and JWH-133, JWH-015, and WIN-55 can all bind to CB2R and elicit an anti-inflammatory response by inhibiting C-X-C motif chemokine 12 (CXCL12) induced chemotaxis and migration of CD4+ and CD8+ T-cells [105,109]. Additionally, cannabinoids can equally affect the nature of cytokines secreted by T-cells. CB2R activation by AEA on CD4+ and CD8+ cells elicit the inhibition of TNF, IFN-γ, and IL-17 secretion. This inhibition of IFN-gamma, a key Th1-polarizing cytokine, consequently, decreases the Th1/Th17 response, a cellular mediated pro-inflammatory response that is heavily implicated in pathogen defense [106,110]. Modulation of T-cell cytokine activity by cannabinoids can play an important role in exacerbating immunodepression, since affected T-helper subsets are integral in modulating the immune response. Following activation, T-cells largely direct the innate immune response and aid in the maturation and activation of B-cells. Like on T-cells, AEA has been shown to decrease mitogen-induced proliferation and apoptosis of B-cells [111]. CB2R activation has been determined to retain mature B-cells in the bone marrow [112].

On the other hand, CB1R activation on macrophages has been found to have mainly pro-inflammatory effects, whereas CB2R activation has been found to have anti-inflammatory effects by reducing ROS production, reducing the secretion of TNF-alpha, and promoting the secretion of IL-10 [99]. This is a shift that can possibly cause a breakdown in defensive measures against pathogens. It must be mentioned that the effects of the endogenous CB1/2R agonist AEA on macrophages appear to be dependent upon the duration of exposure and dose [113]. Chronic exposure to AEA is associated with increased macrophage migration and is thus pro-inflammatory; thus, care must be taken when determining an effective dose for CIDS therapy [109]. Regarding synthetic cannabinoids, JWH-015 and HU-308 are examples of synthetic cannabinoids that can bind CB2R and reduce the migration of monocytes; JWH-015 prevents monocyte migration by downregulating chemokine receptor type 1 and 2 (CCR1 and CCR2) on the monocyte surface and thus desensitizing it to chemokine (C-C motif) ligands CCL2 and CCL3; and HU-308 attenuates monocyte migration by reducing the expression of adhesion molecules and monocyte chemoattractant protein [114,115,116]. This may render monocytes incapable of fighting off an infection and could exacerbate the risk of secondary infections. Collectively, CB2R activation on cells of the innate and adaptive immune system induces cellular changes contributing to immunodepression, therefore increasing susceptibility to infections. This risk must be kept in mind when considering cannabinoids for neuroprotection.

## 6. Conclusions

The endocannabinoid system is deeply intertwined within a multitude of cells, tissues, and organs, presenting a promising frontier for therapeutic targeting. Central to this system are the CB1 and CB2 receptors, which are notably ubiquitous throughout the body. Their widespread presence can potentially complicate targeted interventions, as their influence spans a diverse range of physiological processes. Though the system’s pervasive nature suggests a myriad of potential therapeutic benefits, we are only beginning to decipher its intricacies. With signaling mechanisms and effects still being unveiled, the field offers vast opportunities, albeit with inherent challenges. This review aims to highlight the reported beneficial outcomes, delve into the complexities posed by the ubiquity of CB1 and CB2 receptors, and draw attention to associated risks and considerations. As we navigate this promising realm, a balanced approach combining optimism with caution is paramount. 

## Figures and Tables

**Figure 1 ijms-24-16728-f001:**
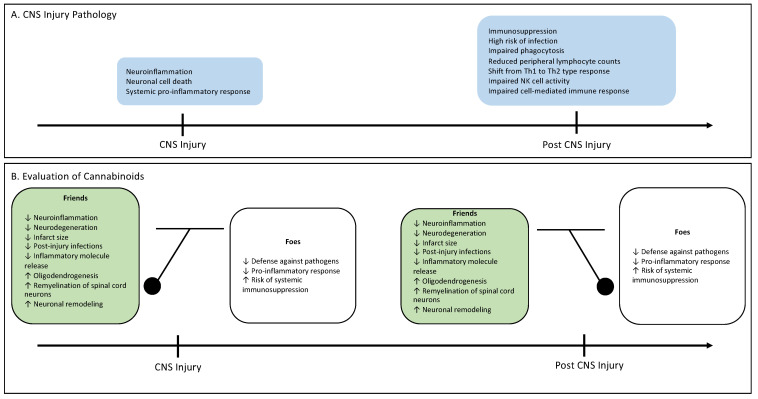
(**A**) A summary of the progression of CNS injury from local inflammation to systemic immunosuppression. (**B**) A pendulum that swings between the benefits and harms of cannabinoid therapy across time.

**Table 1 ijms-24-16728-t001:** Summary of research on the effects of different cannabinoids in the context of different stroke models.

Cannabinoid	Receptor	Model	Effect	Ref.
WIN55,212-2	CB1/2	Chronic cerebral hypoperfusion in rats	Downregulation of TNF-α, IL-1β, -and iNOS	[67]
WIN55,212-2	CB1	Ischemia/re-perfusion injury in rats	Reduced microglial activation	[51]
WIN55,212-2	Partial CB1	Focal cerebral ischemia in rats	Promotes NG2^+^ cell proliferation in ischemic tissue in the brain post stroke	[68]
WIN55,212-2	Off-Target	Human astrocytes (in vitro)	Inhibition of IL-1β-induced inflammatory response	[69]
JWH-015	CB2	Human cell lines LPS- and IFN-G-induced neuroinflammation	Inhibited secretion of IL-1β and TNF-α of microglia	[70]
JWH-015	CB2	Hemi-cerebellectomy in rats	Reduced neuronal cell death, reduced cytochrome-c release	[71]
JWH-015	CB2	Spinal cord hemi-section in rats	Increased bcl-2/bax ratio (reduced apoptosis), reduced cytochrome-c release	[72]
2-AG	CB1/2	Virally induced MS model in mice	Increased IL-4/IL-13 and shift to anti-inflammatory microglia (M2)	[55]
2-AG	CB1	Closed head injury in mice	Decreased BBB permeability and inhibited TNF-α, IL-6, and IL-1β expression following closed head injury	[58]
2-AG	CB1	Closed head injury in mice	Reduced NF-κB-p65 phosphorylation, reduced COX-2 expression	[57]
JZL 184	MAGL Inhibition	LPS (I.P)-induced neuroinflammation in rats	Suppresses formation of ROS and TNF-α by elevating 2-AG levels	[73]
JZL 184	MAGL Inhibition	MCAO in mice	Reduced brain edema, infarct size, and expression of TNF-α; increased IL-10 expression	[74]
Anandamide	CB2	LPS-induced neuroinflammation in primary rat microglia	Reduced release of NO by microglia in response to LPS, reduced expression of M1 phenotype	[75]
Anandamide	CB1	Hippocampal neuron cell line HT22	Reduced ROS production and glutathione expression, inhibited Nox2 expression	[76]
ACEA	CB1	Subarachnoid hemorrhage in rats	Reduced oxidative stress, promoted mitophagy	[61]
ACEA	CB1	MCAO in rats	Prevented brain ischemia/re-perfusion injury, reduced apoptosis	[31]

## Data Availability

Data is contained within the article.

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
