# Peer review of "Neuroprotection and Beyond: The Central Role of CB1 and CB2 Receptors in Stroke Recovery"

_ijms, 2023, doi:10.3390/ijms242316728_

Round 1
Reviewer 1 Report
Comments and Suggestions for Authors
In this manuscript, the authors have provided a review on the role of CB1/2R receptors in stroke recovery. The authors have structured the review, discussing the physiological functioning of the receptors, followed by the reports on the individual roles of CB1 and CB2 in neuroprotection and in facilitating stroke recovery.
The authors discuss the role of the ECS in physiology and how the development of agonists and antagonists for CB1/2R enables the study of these receptors and the importance to understand their mechanisms.
The authors then focus on the individual role of CB1R in neuroprotection, and the use of agonist and antagonist in the protective and survival of neurons. They then examine the importance of CB2R and how specific targeting in terms of agonism and antagonism could facilitate stroke recovery and reduction of neuroinflammation. But the possibility of leading to immunodepression. Overall,
1) The authors suggest that CB1/2R could potentially lead to therapeutic targeting.
2) The review is well written and the significance of the studying CB1/2R in stroke is conveyed.
3) The authors could also include a few sentences about the involvement of ECS in angiogenesis, another aspect in stroke research; Possibly include reference Eitan et al, Gene, Volume 878, 20 August 2023, 147585.
Author Response
Dear Reviewer,
We thank you for your encouraging comments. We have re-read the literature and incorporated the recommended reference into our main body.
The intricate mechanisms of neuroprotection and stroke recovery continue to be a pivotal focus in medical research. As we delve into the cellular and molecular intricacies underlying these processes, the significance of interrelated biological systems becomes increasingly apparent. One such system, drawing heightened attention for its multifaceted role in neurology, is the Endocannabinoid System (ECS). The ECS, with its extensive involvement in neural functions and its potential therapeutic implications, offers a compelling avenue for exploration. For example, recently Eitan et al. (2023) explored the impact of the synthetic cannabinoid XLR-11, a non-selective cannabinoid receptor agonist. Their study revealed that treatment with XLR-11 in human brain microvascular endothelial cells (hBMVEs) led to increased expression of key angiogenic factors such as vascular endothelial growth factor (VEGF), angiopoietin-1 (ANG1), and angiopoietin-2 (ANG2). The elevation of these factors implies a role of ECS activation in improving stroke outcome [12]. Angiogenesis is only one reported benefit among many other beneficial effects of ECS modulation. This leads us to a detailed examination of the ECS, particularly focusing on the selective activation of its receptors and their roles in neuroprotection, anti-inflammation and stroke recovery.
Best,
Bashir
Reviewer 2 Report
Comments and Suggestions for Authors
Abstract
It is too early to state “This can lead to suppressed local and systemic immunity, heightening the risk of infections in patients” in abstract section without discussion – authors need to revise it
Authors should revise bold statements like “Traditional preventive treatments, like antibiotics, have been largely un-15 successful, prompting the exploration of alternatives like immunomodulatory therapies.” –“ As the field continues to expand, a harmonized approach that blends 22 enthusiasm with prudence becomes essential”
Authors should consider including statements related with infection in introduction section – which their focus more on the inflammation connected but distinct things.
Keywords
Authors should change the order to articulate their topic.
Introduction
Authors should briefly start discussion about cannabinoid system to make a way for next sections.
CB1 section
Authors should provide more examples about possible use of CB1 in neuroinflammation. Additionally, translational context should be discussed more deeply in both section related with CB1 and CB2. Based on the review translational explanation is limited.
Author Response
Dear Reviewer,
Thank you for your valuable feedback. Below is our point-by-point response to your comments, detailing the revisions made in the manuscript:
-
Abstract
- We have revised the abstract to reflect a more cautious tone. The statement now reads "Research has shown that compensatory immunodepression usually follows and these mechanisms might influence immunity, potentially affecting infection risks in patients. While traditional preventive treatments like antibiotics face challenges, the exploration of immunomodulatory therapies offers a promising alternative"
- The phrase “As the field continues to expand, a harmonized approach that blends enthusiasm with prudence becomes essential” has been removed.
-
Introduction - Statements Related to Infection:
- We have included additional statements in the introduction to clearly delineate the focus on inflammation and its connection to infection. This sets a more precise context for the subsequent discussion.
"Patients with CNS injuries are more susceptible to hospital-acquired infections, particularly nosocomial pneumonia. Research indicates that CNS injury activates microglia, with this change being associated with acute lung injury in neurotrauma rats [10,11]."
- We have included additional statements in the introduction to clearly delineate the focus on inflammation and its connection to infection. This sets a more precise context for the subsequent discussion.
-
Keywords - Order and Articulation:
- The order of the keywords has been changed to better articulate the core topics of our manuscript and enhance its discoverability in academic searches.
Keywords: Endocannabinoid System, CB1 Receptors, CB2 Receptors, Stroke Recovery, Neuroinflammation, Immunomodulation, CIDS.
- The order of the keywords has been changed to better articulate the core topics of our manuscript and enhance its discoverability in academic searches.
-
Introduction - Discussion about Cannabinoid System:
- A brief overview of the cannabinoid system has been added to the introduction. "The intricate mechanisms of neuroprotection and stroke recovery continue to be a pivotal focus in medical research. As we delve into the cellular and molecular intricacies underlying these processes, the significance of interrelated biological systems becomes increasingly apparent. One such system, drawing heightened attention for its multifaceted role in neurology, is the Endocannabinoid System (ECS). The ECS, with its extensive involvement in neural functions and its potential therapeutic implications, offers a compelling avenue for exploration. For example, recently Eitan et al. (2023) explored the impact of the synthetic cannabinoid XLR-11, a non-selective cannabinoid receptor agonist. Their study revealed that treatment with XLR-11 in human brain microvascular endothelial cells (hBMVEs) led to increased expression of key angiogenic factors such as vascular endothelial growth factor (VEGF), angiopoietin-1 (ANG1), and angiopoietin-2 (ANG2). The elevation of these factors implies a role of ECS activation in improving stroke outcome [12]. Angiogenesis is only one reported benefit among many other beneficial effects of ECS modulation. This leads us to a detailed examination of the ECS, particularly focusing on the selective activation of its receptors and their roles in neuroprotection, anti-inflammation and stroke recovery."
-
CB1 Section - Examples and Translational Context:
- We have provided additional examples illustrating the possible use of CB1 in neuroinflammation."The neuroprotective activity of CB1R is closely linked to its ability to inhibit excitotoxic neurotransmission by blunting pre-synaptic glutamate release. Additionally, there is a strong association between CB1R activity and the expression of brain-derived neurotrophic factor (BDNF), a master neurotrophin in the mammalian forebrain. This connection is particularly relevant in the striatum, where CB1R and BDNF, along with their high-affinity receptor TrkB, play vital roles in the survival, generation, and plasticity of medium-sized spiny neurons. The decline in expression of these elements in animal models of neurodegenerative diseases like Huntington's disease (HD) and Parkinson's disease, and their restoration, prevents HD-like neurodegeneration. Intriguingly, experiments in the R6/2 mouse model of HD showed that enforced re-expression of the CB1 receptor in the dorsolateral striatum allowed the re-expression of BDNF and helped rescue neuropathological deficits​ [49]"
- The translational context, particularly in relation to CB1 and CB2, has been expanded to offer a more comprehensive and in-depth analysis. We have included more detailed discussions on the translational potential of these findings, addressing the limited explanation in the previous version.
- CB1: "Excitotoxicity is considered a critical factor in the progression from ischemia to neuronal death. This understanding has led to clinical trials testing various inhibitors of excitotoxicity in stroke patients. Glutamate, primarily acting through the calcium-permeable ionotropic NMDA receptor, plays a significant role in excitotoxicity. Subpopulations of the NMDA receptor can generate different functional outputs, influencing the progression of neuronal damage during ischemic events [65]. These trials aimed to intervene in the mechanistic steps leading to excitotoxicity to prevent stroke damage. Clinical trials of drugs targeting NMDA receptors for stroke treatment have generally failed to show positive outcomes, often worsening mortality, and morbidity instead of improving them. Drugs like Selfotel, Midafotel, Aptiganel, and Gavestinel, while theoretically promising, were plagued by significant side effects such as agitation, confusion, and hypertension, attributed to the widespread neurological functions of NMDARs [66]. These challenges, coupled with a narrow therapeutic window for effective treatment, have led to the discontinuation of many of these drugs from further clinical development. CB1R modulation presents a promising alternative for stroke treatment due to its distinct mechanism of action and potential for reduced side effects compared to NMDA receptor-targeting drugs. Furthermore, CB1R modulation has been linked to the promotion of neurogenesis and the expression of neuroprotective factors like BDNF enhancing the brain's natural recovery processes. This targeted approach, focused on endocannabinoid system components, offers a more nuanced method to mitigate stroke damage, potentially leading to more effective and safer stroke therapies."
- CB2: "In a large randomized clinical trial involving 1216 patients with acute ischemic stroke, DL-3-n-butylphthalide (NBP), a neuroprotective agent already approved since 2002 for ischemic stroke treatment in China, significantly improved functional outcomes at 90 days compared to a placebo when used alongside reperfusion therapies like intravenous thrombolysis and endovascular treatment. The trial, conducted across 59 centers in China, found that 56.7% of patients in the NBP group achieved favorable outcomes based on the 90-day modified Rankin Scale score, compared to 44.0% in the placebo group, without a notable increase in serious adverse events [93]. NBP’s mechanism of action utilizes similar pathways to CB2R agonism. Namely, NBP has been shown to reduce oxidative stress, reduce neuronal cell apoptosis through PI3K/Akt signaling and improve neurogenesis [94].
In a phase III clinical trial, patients treated with edaravone dexborneol within 48 hours of AIS onset showed better functional outcomes at 90 days compared to those receiving edaravone alone, particularly in female patients [95]. The mechanism of action of this combination involves edaravone acting as a free radical scavenger, reducing oxidative stress and neuronal damage associated with stroke. Dexborneol contributes by enhancing the permeability of the blood-brain barrier, allowing for more efficient drug delivery to affected brain tissues. Additionally, dexborneol exhibits anti-inflammatory properties and may protect against cell death pathways. This study highlighted the importance of a narrow therapeutic window when treating ischemic stroke. The optimal time window is believed to be within 12-48 hours of injury [95]. This parallels the findings of time dependent improvement in stroke outcome using the CB2 agonist AM1241.
Put in this context, CB2R modulation can be beneficial in the early post stroke period by acting on varying cytoprotective pathways."
We hope these revisions adequately address your concerns and enhance the overall quality of our manuscript. We are grateful for your insights and guidance in improving our work.
On behalf of the authors,
Bashir